# Vision Based Modeling of Plants Phenotyping in Vertical Farming under Artificial Lighting

**DOI:** 10.3390/s19204378

**Published:** 2019-10-10

**Authors:** Benjamin Franchetti, Valsamis Ntouskos, Pierluigi Giuliani, Tiara Herman, Luke Barnes, Fiora Pirri

**Affiliations:** 1Agricola Moderna, Viale Col di Lana 8, 20136 Milan, Italy; pierluigi@agricolamoderna.com (P.G.); tiara@agricolamoderna.com (T.H.); luke@agricolamoderna.com (L.B.); 2Alcor Lab, DIAG, Sapienza University of Rome, Via Ariosto 25, 00185 Rome, Italy; ntouskos@diag.uniroma1.it (V.N.); pirri@diag.uniroma1.it (F.P.)

**Keywords:** vision based phenotyping, plants growth prediction, vertical farming, LED

## Abstract

In this paper, we present a novel method for vision based plants phenotyping in indoor vertical farming under artificial lighting. The method combines 3D plants modeling and deep segmentation of the higher leaves, during a period of 25–30 days, related to their growth. The novelty of our approach is in providing 3D reconstruction, leaf segmentation, geometric surface modeling, and deep network estimation for weight prediction to effectively measure plant growth, under three relevant phenotype features: height, weight and leaf area. Together with the vision based measurements, to verify the soundness of our proposed method, we also harvested the plants at specific time periods to take manual measurements, collecting a great amount of data. In particular, we manually collected 2592 data points related to the plant phenotype and 1728 images of the plants. This allowed us to show with a good number of experiments that the vision based methods ensure a quite accurate prediction of the considered features, providing a way to predict plant behavior, under specific conditions, without any need to resort to human measurements.

## 1. Introduction

Eating habits have created a demand for food that is available 365 days a year, regardless of season and location of production. Food travels on average 1500 km, and wastes valuable energy to reach the end consumer. Thirty percent of produce is wasted before it arrives to the end consumer, contributing to 8% of global CO${}_{2}$ emissions. The food that does survive the long journey is not fresh and lacks vital nutrients. Increasingly, consumers are demanding local food, rich in nutritional values and without chemical pesticides and that has caused minimal damage to the environment. According to FAO (https://science.sciencemag.org/content/292/5519/1071), 30% of global energy consumption and 70% of freshwater is used for agriculture [1]. In addition, the increase of the global population suggest that agricultural land availability per capita will decrease in the coming years [2].

The increasing global population, coupled with environmental concerns is leading to the development of new forms of agriculture that consume fewer resources and are not dependent on arable land. Vertical farms with artificial lighting (VFAL) grow crops in vertically stacked layers under carefully controlled conditions, using light-emitting diodes (LEDs) that mimic sunlight and out of soil techniques (e.g., hydroponics and aeroponics) where water is recycled. The ability to precisely tailor the growing environmental also leads to increased yields and crop quality in comparison with field and greenhouse production [3].

### VFAL and AI Based Methods

As VFAL does not rely on external environmental conditions and is protected from changes in weather or soil fertility, it enables the production of crops year round with minimal demand for water, nutrients and land [4]. Each farming unit is its own individual ecosystem, creating the exact environment for plants to flourish. By developing the optimal light spectrum, temperature, humidity, pH, and nutrients, researchers can ensure the best possible flavor, color and nutritional quality for each plant.

Nonetheless, the large energy associated with especially lighting but also cooling and dehumidification is limiting the diffusion of VFALs [5]. Consequently in VFAL optimizing resource efficiency, while maximizing yield and quality is critical.

Under these premises, plant growth optimization and quality assessment while minimizing the required resources asks for AI methods to ensure a highly flexible technology for a good rate of development. By capturing many data and images relating to the plant physiology, growth and environmental conditions, it is possible to estimate information of the plant growth behavior.

Accuracy of data collected and predicted by methods based on vision and advanced sensory tools, with respect to physiological measurement, makes it possible to create predictive models of plant productivity based solely on automated data collection. Additionally, discovering relationships between image sensor data and more in-depth parameters of crops such as flavor, nutrient deficiency and disease presence, high-throughput AI based input testing could be developed. Accelerating the screening of inputs to the VFAL system, as well as predictive tools for optimizing environmental inputs are important for enabling VFAL to significantly contribute to reductions in the environmental impact of future food production.

## 2. Related Works

Since commercially operative VFALs are a relatively recent phenomenon, predictive tools for crop quality and yield are not currently available. Conventional field and outdoor agriculture by comparison has benefited from long-standing multidisciplinary and legislative efforts to provide methods for improving resource use efficiency and productivity [6].

Precision agriculture has arisen as a field that uses data science, computer vision (CV) and machine learning (ML) to increase agricultural productivity while minimizing its environmental impact [7]. More recently, precision agriculture techniques have provided tools for improved crop management through detailed monitoring of environmental conditions, maps of spatial variability in soil conditions and both remote and proximal sensing of reflected and radiated energy from crop surfaces [8,9]. Liakos et al. [7] provided an excellent review paper on the ML techniques employed in field agriculture. Specifically, Liakos et al. [7] reviewed studies which used ML methods to model the yield prediction, disease detection, weed detection and crop quality in field agriculture. Yield prediction is critical in precision agriculture as it enables to increase crop management and productivity [7]. For example, Ramos et al. [10] developed non-destructive method that automatically counted coffee fruits and estimated their weight on a branch.

Vision-based analysis of crops are an essential component of remote sensing in agriculture. For example, the advent of hyperspectral cameras, which allow images to be divided into many spectral bands, have led to the development of vegetation indices which can be used to estimate growth parameters including biomass, chlorophyll content and leaf area index [8]. Recently, hyperspectral cameras have been developed which can be attached to unmanned aerial vehicles and used to capture spatial resolutions of up to 1 cm. Consequently, much attention has been given to the development of methods for extracting important information on crop development using hyperspectral imaging. For example, Sylvain et al. [11] showed that vegetation indices extraction from ultra-high resolution images can be used to estimate green fraction, green area index, leaf/canopy chlorophyll content and nitrogen contents of sugar beet crops.

Monitoring of greenhouse crops using remote sensing-type applications also show potential for improving knowledge of crop requirements. In contrast to field-based farming, greenhouses allow greater control of the growing environment. Nevertheless, the practical application of remote sensing for yield prediction in many crops is limited.

With the advent of new high-quality cameras and the recent advances in CV due to deep learning methods, RGB and RGBD images have played a relevant role too. Ballesteros et al. [12] demonstrated that onion yield predictions can be made using RGB images which are processed to determine plant height and volume using a modified version of Volume and Leaf Area Index Calculation (VOL-LAIC) software.

However, monitoring often focuses entirely on the macro-environmental parameters, which is only partially related to the microclimate surrounding the leaves. Story and Kacira [13] developed a crop monitoring system which captures and processes RGB, NIR and thermal image data using a machine vision system to determine overall plant growth and health, as well as parameters for plant phenotyping applications.

Nevertheless, CV applications in greenhouses are made difficult by several sources of variation, e.g., variations in solar intensity, an issue that is likely to be compounded in facilities using variable-intensity supplemental lighting. Since VFALs provide a greater degree of environmental stability than either field or greenhouse cultivation, the development of intelligent crop monitoring systems will enable very high-resolution crop management.

## 3. Contribution and Overview

### 3.1. Contribution

The aim of this study was to show how, using AI and specifically vision based methods, it is possible to obtain information on the plant phenotype in a closed VFAL environment. A major difficulty in plant phenotyping is to document day by day the plants response to several factors even in controlled environments such as vertical farms.

Consequently, there is still a lack of experimental studies evaluating the automatic measurements made possible by novel techniques with ground truth taken with manual measurements. Manual measurements require a punctual and meticulous and continuous effort to process a great amount of data along the growing cycle of a plant.

This work provides a joint effort between the agronomy and computer science fields to evaluate the effectiveness of off the shelf research methods, suitably adapted to the plants conditions, for some relevant aspects of phenotyping. Relevant data were collected digitally, visually and manually to both facilitate the development of the training set and provide outputs from the model, which are crucial for fast development of VFAL technologies.

The authors performed various experiments on the growth of Basil, where a large dataset was collected both manually and visually for the first time. Specifically, through three cycles of experiments, 2592 data points were manually collected on: (1) plant height; (2) leaf weight, for the whole set of leaves of each plug; (3) number of leaves per plant; and (4) weight of each plug. In parallel, 1728 images were collected. To the best of the authors knowledge, this is the first time many data were gathered to compare predictions and observations, and the models validated.

The main outcome of the research is the identification of relevant phenotype features with images and precise computation of three important growth measures, namely plants heights, leaf area and leaf weight, at different stages of growth (see Figure 1). Precise here means that the accuracy of the prediction could be assessed by comparing the results with manual measurements. These results are reported in Section 5.

### 3.2. Paper Outline

In the following sections, we describe each of the above contributions providing results against collected ground truth data in Section 5.

In Section 4, we introduce the vision and learning methods leading to the above results. First, in Section 4.1, we introduce the outcome of image acquisition, the noise removal and 3D modeling. Further, in Section 4.2, we explain the approach taken to measure plants height at specific grow days, Days After Seedling (DAS), namely 7, 18, 21, 24, 28, based on three-dimensional depth maps and possible alternative methods based on point clouds. After introducing the computation of height, in Section 4.3, we introduce the processing required to measure the leaf area, restricted only to the surface (i.e., visible) leaves, without losing generality since it turns out to be exactly the same process approach used when taking manual measurements. In addition, for leaf area, we provide accuracy and derive the leaf weights according to the density computed by the experimental studies and described in Section 5.

Finally, in Section 5, we illustrate the results and evaluate accuracy against the ground truth observed measurements taken by the agronomists.

## 4. Vision Process Flow

As described in more detail in Section 5, each experiment was carried out for 28 DAS, for the six layers shown in Figure 2.

Our task was to predict the height of the plant from DAS 7 to DAS 28 and to predict both the leaf size of the visible layer of leaves for the last DAS (DAS 21 to 28) and leaf weight.

### 4.1. Acquisition and Noise Removal

To collect images for the entire DAS cycle, we positioned commercial close range RGBD sensors [14], one for each rack layer, at about 54.3 cm from the rack plane and at 50.5 cm from tray plane, as illustrated in Figure 2. The cameras were fixed at 50.5 cm from the plant substrate and 54.5 cm from the bottom of the trays, because this distance represents the typical distance between the LEDs and plant base in a VFAL. Each high density tray contains 144 plugs and each low density tray contains 104 plugs. Since each frame images two trays, each image collects either 144×2 plugs for the high density layers or 104×2 plugs for the low density layer.

An RGBD image realizes two representations: a color image with three color channels (R, G and B), returning a tensor of size H×W×3 and a depth map with only the depth values, namely each pixel at the *x* and *y* coordinates of the image specifies the depth value *d*, given in mm, indicating how far away the imaged object is.

The RGBD sensor has intrinsic parameters denoted by a matrix *K*, specifying the sensor focal length *f*, the scale values $m_{x}$ and $m_{y}$ specifying the number of pixels per mm., a skew parameter *s* and the center of projection $\left(p_{x},p_{y}\right)$, hence:
(1)$$K=\left(\begin{array}{ccc} fm_{x} & s & p_{x}m_{x}\\ 0 & fm_{y} & p_{y}m_{y}\\ 0 & 0 & 1 \end{array}\right)$$

The intrinsic matrix allows projecting a point in space measured in mm into a point on the image plane, measured in pixels, and to deproject a pixel on the image plane to a point in space. Together with the intrinsic *K* distortion parameters are made available for radial corrections $\left(\kappa_{1},\kappa_{2},\ldots \right)$ such that the corrected coordinates $\left(\hat{x},\hat{y}\right)$ of a pixel are obtained as: $\hat{x}=x_{c}+L\left(r\right)\left(x-x_{c}\right)$ and $\hat{y}=y_{c}+L\left(r\right)\left(y-y_{c}\right)$, where $\left(x_{c},y_{c}\right)$ is the distortion center, (x,y) are the distorted coordinates, and $L\left(r\right)=1+\kappa_{1}r+\kappa_{2}r+\ldots$, with $r^{2}={\left(x-x_{c}\right)}^{2}+{\left(y-y_{c}\right)}^{2}$ (see [15]).

Typically, RGBD sensors require a procedure to align the RGB image to the depth image, which is also useful to obtain a point cloud from the depth map with points colored as in the RGB image. To map a depth image to a point cloud the usual formula, with *d* the depth, R a rotation matrix and t the translation vector, is:
(2)$$\begin{array}{c} \left(\begin{array}{c} X\\ Y\\ Z \end{array}\right)=dK^{-1}R\left(\begin{array}{c} \hat{x}\\ \hat{y}\\ 1 \end{array}\right)+t \end{array}$$

*X*, *Y* and *Z* are the coordinates of the scene with the camera as the global reference point, while $\hat{x},\hat{y}$ and *d* are the corrected coordinates of the depth image.

Because the camera is fixed, a single image is not quite dense, and there are points of reflection where the sensor does not return data. This can be observed in Figure 3, where RGBD results without specific processing are shown. In particular, we can note in Figure 3c the holes in the depth map, and in Figure 3d that the produced point cloud is noisy. To address these problems and obtain an almost dense representation, we merged multiple consecutive depth maps and smoothed the resulting image with a Gaussian kernel. This merging makes sense, even though the camera is fixed, since natural flickering of the plants illumination and the ventilation, moving the leafs and their illumination, changes the image of the leaves, thus have the effect of small camera motions (see Figure 4a).

In fact, because the point cloud results from the depth images, all taken from the same vantage point, as the camera is fixed (see Figure 2), noise elimination is not trivial and may result in removal of useful points. To reduce the noise, we first fitted a plane to the point cloud, parallel to the tray surface, thus eliminating all points beyond the plane (see Figure 4b,c). Further, we used the well known Iterative Closest point (ICP) algorithm [16,17], exploiting two point clouds from merged depth maps to clean points with loose correspondences to the common fitted plane.

Finally, the point cloud was rescaled using a known metric of the imaged objects, namely the dimension of the cell where plugs are put in, and the tray dimension. This step is required for two reasons: first because the camera, despite being fixed, is not well aligned with the tray plane, and secondly because the intrinsic parameters mapping pixels to mm are not sufficiently accurate. To rescale, we considered four points, which are the four vertices of a visible cell on the tray, namely the cell where plugs are located, which we know are separated by 30 mm, and computed the transformation T mapping the current points to the expected ones. The transformation T is an affine transformation including rotation, translation and scaling, such that: $V_{E}=TV_{M}$, where $V_{E}$ is the expected matrix formed by the four points in real world dimension, and $V_{M}$ is the matrix of points as measured on the point cloud. According to the transformation, effective measures can be taken on the point cloud also at close distance, as illustrated in Figure 4d.

We actually used the depth map for the height prediction (next section) and the point cloud for the leaf area and weight prediction.

### 4.2. Height Prediction

Given the basic setting described above, we next elaborate about automatic plant height computation at each defined DAS step.

A relevant insight given by the agronomists in height computation is the difference amid plants at growth stages, namely DAS, according to plant density (see Section 5.1). This information can be hardly assessed by the average growth for each tray and the heights standard deviation. In fact, it is interesting to note, for example, that the standard deviation significantly increases at later growth steps and according to density, namely the most dense trays have lower standard deviation because both plants competitiveness and lack of space stabilize plants growth at similar heights. Note that, because there are more than one plants in each plug, the agronomists measure the highest plant. However, it seems necessary to predict the height surface according to the region where each plant grows, namely near the cell on the tray where its plug is located.

To obtain height measures for each tray and each DAS, we considered the depth maps obtained day by day for each rack involved in the experiments. However, the plant surface provided both by the point cloud and the depth map hardly distinguishes individual plants, especially at later stages, at which only a carpet of leaves is visible.

To obtain quite precise measurements for each cell, where plugs are located, we need to recover the cells, which at later stage are clearly not visible.

To this end, we fitted lines to the tray images taken at the early DAS. Lines were computed first computing a Canny edge detector of images of early stages, where the tray cells are still visible. The Canny edge detection is shown in Figure 5, where the image is obviously rectified and undistorted. Then, from the binary image of the edges, the lines were obtained with the Hough transform, and a new binary matrix *L*, with zeroes at the line locations, was obtained. Then, letting *M* be the depth map and *L* be the image of the line fitted, the new depth map with the line fitted is $M_{l}=M\circ L$, with ∘ the pointwise Hadamard product.

The new depth map $M_{l}$ returns the distance between the camera and the plants, except for the lines bounding the cells, where plugs are located (see Figure 5), where as mentioned above the value is zero.

As noted above, the *x* and *y* coordinates are given in pixel. In other words, the depth map is a surface covering an area of 720×1280 pixels, while the tray area measures 110 cm × 63 cm. However, since the depth *d* is measured in millimeters, computing the height rescaling is not required.

Finally, the plants height surface was obtained by subtracting from the depth map the distance to the plane parallel to the tray surface, which is known to be 505 mm, as noted above.

Let D(π,τ) be the distance between the camera image plane π and the surface plane τ of the tray, measured along the ray normal to the camera image plane, and let $M_{l}$ be the depth map, with lines bounding the cells. The depth map was smoothed by convolving it with a Gaussian kernel of size 11×11 with σ=1.3. Then, the plant height surface $H_{p}$ was computed as:
(3)$$H_{p}=D\left(\pi ,\tau \right)-M_{l}+\epsilon$$
with ϵ Gaussian noise, possibly not reduced by smoothing.

To compute the height of plants relative to a plug, we only need to consider all the points within the cell bound by the appropriate vertices. Indeed, we can just consider the projection on the plane of the surface points, as far as the *z* coordinate (note that we name it *z* because it is the plant height, not the depth *d*) is not zero. First, note that *V* is a vertex on the tray lines if all its four-connected points are equal to 0. Therefore, let $V=\{V_{j}|V_{j}$ be a vertex of the approximate rectangle bounding a tray cell, j=1,…,n}. Let $\hat{p}=\left(x,y\right)$ be the projection on the plane of a point $p={\left(x,y,z\right)}^{⊤}$ on the surface height, z>0. Then, $\hat{p}$ is within a cell *q* bounded by the vertices $\langle V_{i,q},V_{j,q},V_{k,q}\rangle \in V$ iff the vertices are connected and:
(4)$$\begin{array}{ccc} L(p,q) & = & \left(\left({\left(V_{i,q}-V_{j,q}\right)}^{⊤}\left(V_{i,q}-V_{j,q}\right)\right)>{\left(V_{i,q}-V_{j,q}\right)}^{⊤}\left(V_{i,q}-\hat{p}\right)>0\wedge \left({\left(V_{j,q}-V_{k,q}\right)}^{⊤}\left(V_{j,q}-V_{k,q}\right)\right)>\right.\\ & & \left.({\left(V_{j,q}-V_{k,q}\right)}^{⊤}\left(V_{j,q}-\hat{p}\right))>0\right)\equiv ⊤ \end{array}$$

Then, the set of points on the height surface of a cell *q* is $H_{q}=\left\{p={\left(x,y,z\right)}^{⊤}|L\left(\hat{p},q\right)\equiv ⊤\right\}$. This defined, it is trivial to determine the highest and average height for each cell *q*, and the height variance within a cell:
(5)$$\begin{array}{c} h_{max}\left(q\right)=\max_{z}\left\{\left(x,y,z\right)\in H_{q}\right\},\\ h_{\mu }\left(q\right)=\frac{1}{N}\sum_{{\left(x,y,z\right)}_{j}\in H_{q}}^{N}z_{j}\\ \sigma_{q}=\frac{1}{N-1}\sum_{{\left(x,y,z\right)}_{j}}^{N}{\left(z_{j}-h_{\mu }\right)}^{2} \end{array}$$

Here, $N=|H_{q}|$, with |·| the set cardinality.

The results and comparison of the predicted height with the measured ones, according to the modeled surface, are given in Section 5.

### 4.3. Leaf Area and Weight

In this section, we discuss the computation of both leaf area and leaf weight. We recall that leaf area was not measured by the agronomists, however they collected the weight of the leaves of the plants in a plug, and the number of leaves. Furthermore, they made available, from previous studies, a sample of leaves for the considered species with area and weight. From this sample, they measured $\rho_{BL}=\frac{W_{BL}}{Area_{BL}}$, where $W_{BL}$ is the weight of a basil leaf, $Area_{BL}$ is the area of the leaf and $\rho_{BL}$ is the density. From this sample, the weight, and the knowledge of the number of leaves, we derived a reference ground truth for leaf area to be directly comparable with our predicted leaf area.

From the predicted leaf area, using the density, we computed the leaf weight. However, we could not predict thus far the number of leaves in a plant or the number of plants in a plug.

For predicting the leaf area, we proceeded as following. We computed instance segmentation of superficial leaves at DAS 18, 21, 24 and 28. Using the colored mask and Equation (Equation 2), we projected the mask to the point cloud to obtaini a surface well approximating in dimension and shape the considered leaf. Because each point of the projected mask is colored, the mask can also be segmented from the point cloud, simply selecting all points with the specific color. Note that the point cloud can be saved as a matrix of dimension N×6 where *N* is the number of points, the first three column are the point locations and the last three the associated color.

Then, we fitted a mesh surface to the chosen leaf, eliminating all the redundant meshes. The process is shown in Figure 6 and Figure 7.

As noted in Section 3, we can measure leaf area only for the surface leaves not occluded by more than 60% by other leaves. To obtain this partial yet significant result, we resorted to leaf segmentation with Mask R-CNN [18], a well known deep network for instance segmentation. We used our implementation in Tensorflow, which is freely available on GitHub (See https://github.com/alcor-lab).

As is well known, CNNs typically require a huge collection of data for training, and in our case well known datasets such as COCO (https://github.com/cocodataset/cocoapi) could not be used. In addition, we noted that leaves completely change aspect after the early DAS, namely around DAS 17, when they gain a characteristic grain.

Therefore, we decided to collect both small and large leaves of almost mature plants, starting from DAS 17.

We collected only 270 labeled large leaves and 180 small leaves, with LabelMe (See http://labelme2.csail.mit.edu/Release3.0/browserTools/php/matlab_toolbox.php) and following Barriuso and Torralba [19], selecting images from Rack 7, the first experiment and DAS 17, 18, 19, 21, 23, 24, and 28. We considered only Rack 7, to make testing on the remaining racks clearer. However, in the end, we also used Rack 7 for testing, although we delivered separate tests to not corrupt results.

We tested the network on two resolutions, as shown in Figure 6, obtaining very good results for both training and testing. The accuracy of each large leaf recognition is given within each bounding box; note that small leaf accuracy is not shown.

It is apparent that, because the number of samples is so small, leaves occluded by more than 60% of the area are not recognized. However, we noticed that the samples we could collect from the rack surface are good representatives of the whole leaves set. Furthermore, as explained in the sequel, this does not affect significantly the measured area.

Given a leaf mask, as obtained from the RGB images, first, it was projected on the depth map, as shown in the lower panels of Figure 6 and, further, using Equation (Equation 2), the points of the point cloud were obtained [20,21]. Note that the projection of a depth map point (x,y,d) into space coordinates (X,Y,Z) allows recovering the colors as well. To maximally exploit the segmentation masks for each image, we assigned a different color to each mask, so that when two masks were superimposed it was easy to distinguish to which mask leaf points belong. More precisely, the color masks were projected into space together with the points on the plants surface height (the depth map) and, because each mask was assigned a different color, we could use them as an indicator function:
(6)$$\begin{array}{c} P=\left(X,Y,Z\right)\in Leaf_{j}\equiv \left\{\begin{array}{c} 1\;\mathrm{if}{\left(R,G,B\right)}_{j}=\left(0,1,0\right)\\ 0\;\mathrm{otherwise} \end{array}\right. \end{array}$$

Note that, to exemplify, we assumed that the mask for leaf *j* is green.

Since we used more than a depth image, thereby generating more than one point cloud, for the leaves occluded on the RGB image there are two possibilities. Either the occlusion remains the same in space or it can be resolved; however, the mask does not extend on the parts which have been disoccluded in space. In this last case, to recover the occluded part of the leaf not distinguishable by the lack of the segmentation mask, we resorted to the computation of normals using *k*-nearest neighbor and fitting a plane to close points. As is well known, for three points on the plane, the normal is simply obtained as the cross product of the vectors linking them, further normalized to a unit vector. Given points ${\left(P_{1},\ldots ,P_{n}\right)}^{⊤}$ on the boundary *b* of a mask, the boundary is moved according to the normals to close by points $P_{j}^{\prime }$. Normals are assigned a cost according to the distance to the points on the boundary and according to the cosine angle between the normal on the boundary and normal beyond it:
(7)$$\begin{array}{c} w_{n}\left(P,P^{\prime }\right)=\left(\kappa D\left(n_{P},n_{P^{\prime }}\right)+\left(1-\kappa \right)\left(n_{P}\cdot n_{P^{\prime }}\right)\right)\\ w_{P^{\prime }}^{⋆}=\arg\min_{w_{n}}\left\{w_{n}\left(P,P^{\prime }\right)\;|\;P\in b,P^{\prime }\notin b\right\} \end{array}$$

The above essentially assigns a cost $w_{P^{\prime }}^{⋆}$ to each point $P^{\prime }$ beyond the boundary of a leaf mask, by choosing only those points whose cost of the normal is less than a threshold obtained empirically, hence it allows to extend, when this is possible, the regions of occluded leaves, without significant risks.

Having obtained the segmented leaves in space, and recalling that the space is suitably scaled so that distances correspond to real world distances, from the transformation of pixels into mm, we could measure the leaf area, even though in many cases the leaf was not full, because of occlusions, as noted above.

To obtain the leaf area, we built both a triangulation and a quadrangular mesh. The triangulation, despite being more precise, is more costly and the triangles do not have the same shape and size, therefore area computation requires looking at each triangle. The triangulation can be appreciated Figure 7a.

To fit the quadrangular mesh to the leaf points, we used the spring method (see [22]) solving the fitting as a non-linear least square problem, with relaxation.

The results of the fitting for the quadrangular meshes are given in Figure 7b.

Finally, we checked if a mesh is occupied by one or more points, and if it is not, then the specific mesh was removed, but only if it is on the boundary. We can see in Figure 7 that the result is sufficiently accurate, for measuring area, clearly not for graphical representations, and it is quite fast. The only difficulty is the choice of the size of the quadrangles, which determines the smoothness of the shape and, obviously, the goodnesses of fit, yet augmenting the computation cost.

Hence, the leaf area is simply: $\sum_{i=1}^{N}A_{i}$, where each $A_{i}$ is determined by the choice of the size of the quadrangular meshes.

The results on the accuracy of the computation are given in Section 5. To obtain the leaf weight, we simply used the above density formula, and obtained the weight multiplying the recovered area with the density $\rho_{BL}$. For the weight, we show the accuracy in Section 5.

## 5. Results

In this section, we present the results of our experiments. First, we recall how the experiments were conducted at the farm, and then we consider the specific features predicted for automatic phenotyping, namely, plant height, plant leaf area and leaf weight. Finally, we shortly address feature discriminant analysis for the high and low density of plants.

### 5.1. Experiments Method

The experiments analyzed the growth behavior of three species of basil, namely Ya, Yb and Yc, and their vital cycle confined within 28 Days After Sowing (DAS).

Each experiment was performed on two racks, as shown in Figure 2, one for high density and the other for low density plants distribution. High density implies 2574 plants per m${}^{2}$ (144 × 2 plugs per layer) and low density implies 1872 plants per m${}^{2}$ (104 × 2 plugs per layer). Each rack consisted of three layers, one for each variety of Basil, were the racks are numbered from 7 to 12. Racks 7–9 are for low plants density and Racks 10–12 are for high plants density.

Each layer was equipped with a camera, and set of sensors including: water and air temperature, humidity, $CO_{2}$, PH and EC. The data from the sensors were collected with a Data Acquisition System (DAQ), which controls if there are any significant variations with the environmental conditions.

The LED lights used were 28 W Valoya A673L, with the following spectrum characteristics: 14% blue, 16% green, 53% red and 17% far red. The LED lights were positioned such to provide the necessary light intensity varying between 200 and 240 μ mol/m${}^{2}$.

The overall conditions throughout the experiments are shown in Table 1. The nutritions in the tank were controlled using a PH and EC meter and adjusted manually.

Each experiment considered three phenotype features: plant height, leaf area and leaf weight. The objective was to provide accurate measures of these features, given variability of only a single condition, in particular here the single variable condition is plant density. All other factors affecting the features, such as light, water, nutrients, substrate, humidity and ventilation are assumed to be the same for all plants, as shown in Table 1.

An experiment consisted of two parallel monitoring techniques: manual monitoring, with measures obtained by systematically harvesting group of plants from each tray, and vision based monitoring, with measures automatically derived for all DAS.

The measures taken manually were: (1) plant height; (2) leaf weight, for the whole set of leaves of each plug; (3) number of leaves per plant; and (4) weight of each plug. The height was measured as is (i.e., the plant was not manually elongated to measure the full height). For the remaining measurements at DAS 21 (gray), DAS 23 (blue), DAS 25 (yellow) and DAS 28 (green) (see Figure 2 right), destructive testing was performed by cutting different sections of the layer.

In total, 2592 manual measurements for the mentioned phenotype features were recorded for three experimental cycles.

Visual monitoring started at DAS 7, when plants were moved on to the racks, and continued up to DAS 28, and was based on four images taken each day for each rack with a commercial RGBD camera, as described in Section 4.1. In total, 1728 images were collected for the three experimental cycles.

### 5.2. Height Prediction

Height was measured starting at DAS 7 and height vision monitoring started at the same time. As described in Section 4.2, height was measured at each point of the depth map, with points within a cell collected together to have a more precise monitoring of plants growth in a specific region.

Figure 8 shows the height measured as described in Section 4.2. For visibility purposes, cells are grouped into a neighborhood of four cells, and we excluded the first row for the low density and the two extreme columns for the high density racks.

Note that for these precise vision based measurements of the height, for all the 21 DAS (7–28), we do not have the same manual measurements. Manual measurement of the whole rack would have required to harvesting all the plants.

Comparison with manual measurements are given in Figure 9. Here, to assess the error between the predicted and measured plants height for each rack, we considered the mean and variance of the retrieved values, and computed the Bhattacharyya distance. Note, in fact, that, while we have the highest value of about 40 plugs manually measured for DAS 18, 21, 24, 25, and 28 and rack, we have about (i.e., excluding the zeroes on the fitted cells separation lines) 720×1280 measured points from the depth maps, therefore a distance between probabilities is more convenient and correct. Given $\sigma_{p}$ and $\mu_{p}$ are the standard deviation and mean of the predicted values *p*, for each DAS and rack, and similarly given that $\sigma_{m}$ and $\mu_{m}$ are the standard deviation and mean of the manually measured values *m*, for each rack and DAS, the distance between the two height measures, at each DAS and rack, is defined as:
(8)$$D_{B}\left(p,m\right)=\frac{1}{4}\log\left(\frac{1}{4}\left[\frac{\sigma_{p}^{2}}{\sigma_{m}^{2}}+\frac{\sigma_{m}^{2}}{\sigma_{p}^{2}}+2\right]\right)+\frac{1}{4}\left(\frac{{\left(\mu_{p}-\mu_{m}\right)}^{2}}{\left(\sigma_{p}^{2}+\sigma_{m}^{2}\right)}\right)$$

Clearly, $D_{B}$ is an error distance between the predicted and the measured values.

There are interesting observations to make. First, note that plants height reaches higher values where plants are more dense, and the standard deviation is slightly inferior, because of competitiveness amid plants. This is somewhat expected as increased density means that plants will compete for the photons from the LEDs. In Figure 10, assuming height is normally distributed, we provide a comparison between the probability density functions (pdf) generated by the two measures.

From the above results, as expected, it turns out that height measured from the depth maps is extremely accurate. The slight difference is due to the greater amount of data on the whole surface, collected from the vision based method, as opposed to the manual method.

### 5.3. Leaf Area and Weight Prediction

As already highlighted in Section 4.3, leaf area was not measured by the agronomists and only the weight was measured. However, several samples of basil leaves from the three species were taken and both weight and area for these samples were measured. Furthermore, note that weight was given for the set of all leaves for a plug, and the number of collected leaves was also given.

To properly confront the predicted leaf area with something derived by the true measurements, we proceeded as follows. First, instead of considering the density $\rho_{BL}=\frac{W_{BL}}{Area_{BL}}$ (see Section 4.3), we computed a nonlinear regression using the few samples of basil leaves about which we had both the weight and the leaf area.

We resorted to a fully connected network with two layers and obtained the parameters, so that for batches of values of the same size as the tiny training set we obtained a prediction of a leaf area, given its weight. Note that the accuracy of the net, to predict leaf area given leaf weight, is 100% on the training set, while for example with a linear regression we obtained 20% accuracy on the training set.

Once the relation between single leaf weight and single leaf area was assessed, via the fully connected network, we used the total leaf weight computed for each of the harvested plugs, at specific DAS, and the number of leaves for that plug, to obtain a distribution of leaf area per rack at DAS 21, 24, 25 and 28. These values were used for comparison with the vision based methods.

More precisely, let net be the fully connected network function predicting the area given a weight *w* of a single leaf, and let $N_{\ell }$ be the number of leaves for the specific plug, then the leaf area predicted by the weights is (see Figure 11):
(9)$$A_{W}=net\left(\frac{1}{N_{leaves}}W_{leaves}\right)$$
where $W_{leaves}$ is the weight of all leaves collected for a plug and $N_{leaves}$ is the number of leaves collected for that plug. In particular, as repeated several times, the manually collected data are available only after harvesting the plants.

The leaf area obtained by the vision method is defined as $A_{VM}$, and the method is described in Section 4.3, and is available as an average measure (based on the segmented leaves for image), for all DAS.

A comparison between the two computed leaf areas is given in Figure 11. We can note from the figure that, where there are values for the weights (at DAS 21, 24, 25, and 28), the difference between predicted area from vision and predicted area from weights is minimal or even not existent, while there is a large difference where there are no reported values for the manually collected weights, and weights are simply fitted.

This implies that either both the net and the vision based area inference provide the same wrong measurements or that both are quite accurate.

To verify the accuracy of leaf area prediction, we computed the leaf weights from the predicted leaf area from vision, by inverting the network, namely we used the network output to compute back the weights. Of course, this is subject to noise and prone to errors. The results on the recomputed weights are given in Figure 12. We can see that again the error is quite limited where the predicted weight (from vision based predicted area) are compared with manually collected leaf weights, at DAS points where the values were measured, while it is relatively high where they are simply fitted in the lack of provided values.

The fitting of unknown values was based on polynomial fitting.

Note that for the weights we also computed the error using the Bhattacharyya distance as for plants height, for the same reason given for the height, namely there are no comparable samples, although there are comparable probabilities from these samples.

### 5.4. Discriminant Feature Analysis for High and Low Plants Densities

A final observation on the reported results concerns the features chosen. Plant height, leaf area and leaf weight are typical features for plant phenotyping. A typical question is: Do these features discriminate between high and low plant densities?

We showed that the plant height distribution, namely the mean and variance, actually highlight the difference, because the variance is slightly inferior for high density. Here, we show in Figure 13 that also the area combined with the weight discriminates between high and low plant density, while for weight this is less evident using combined features. It is clear, however, that the low density leaves have higher weight, from the histogram shown in the last panel of Figure 13.

### 5.5. Implementation

All algorithms were implemented in Python 3.6, Matlab 2019a, and Tensorflow. We used a i9CPU with 2.9 GHz, 128 GB of RAM and 4 NVIDIA Titan V. For the point clouds, we used PCL, CloudeCompare and MeshLab.

## 6. Discussion and Conclusions

This work proposes an automatic method for extracting phenotype features, based on CV, 3D modeling and deep learning. From the extracted features, height, weight and leaf area were predicted and validated with ground truths obtained manually.

This study is particularly significant because it exploited many data collected with extreme precision and due diligence. Manually collected data made it possible to evaluate the accuracy of the vision methods.

This, thus far, seems to be a unique kind of experiment in phenotyping, being based only on RGBD images. The results show that the plant height, leaf area and weight obtained using inexpensive RGBD cameras matched closely with the detailed measurements.

These results are especially relevant in indoor VFAL conditions, where plants are stacked vertically and in difficult to reach conditions.

The ability to obtain detailed information on the plant weight and therefore yield, without employing destructive techniques, facilitates the process of automation of the growth of vegetables in indoor VFAL conditions and consequently can substantially diminish the costs of production. In addition, by gathering a large amount of image based information on the plant growth and phenotype, it is possible to train the validated algorithm to provide information which aids the optimization of plant growth and therefore the reduction of resource use.

More importantly, this study shows how visual based monitoring methods, which are gaining increased attention in outdoor, “traditional” agriculture can be adapted and successfully used for indoor VFAL. In fact, the authors believe that the ability to closely control the VFAL growing environment and input conditions makes visual based methods especially useful for monitoring and optimizing the growth of fruits and vegetables in indoor VFAL.

This study opens the way for future work on the ability to obtain quantitative information on the plant phenotype and eventually determine qualitative information such as nutrition values and taste using vision based techniques in a closed and closely monitored VFAL environment. Specifically, it is possible to understand the effect of LED light (e.g., intensity, spectrum, and cycle), environmental conditions (e.g., CO${}_{2}$, humidity, temperature, and PH) and grow conditions (e.g., plant density, seed type, nutrients, etc.) on crop growth behavior. The two main objectives were: (1) to maximize growth and plant quality while minimizing the resource use; and (2) determine nutritional other qualitative values (e.g., taste) and the presence of external agents on the crops using inexpensive and visual based methods rather than expensive and time-consuming non-destructive techniques.

## Figures and Tables

**Figure 1 sensors-19-04378-f001:**
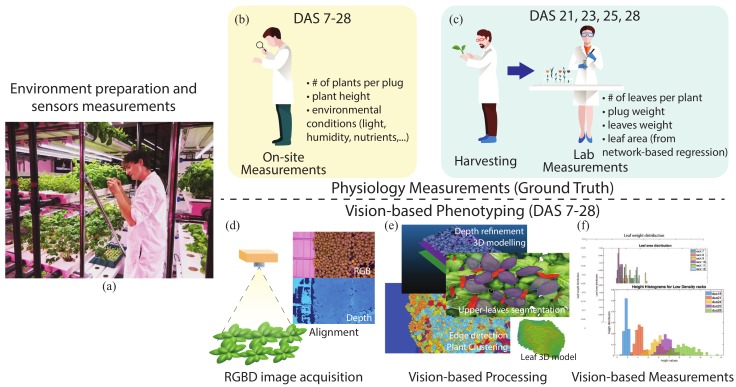
The figure shows the schema of the two parallel computational processes leading to automatic measurements: the experiment setting (**a**); the manual data collection via on-site (**b**) and lab measurements (**c**); the images acquisition and alignment (**d**); and the ML and CV methods (**e**) to predict phenotype features, such as plants height, leaf area, and leaf weight (**f**).

**Figure 2 sensors-19-04378-f002:**
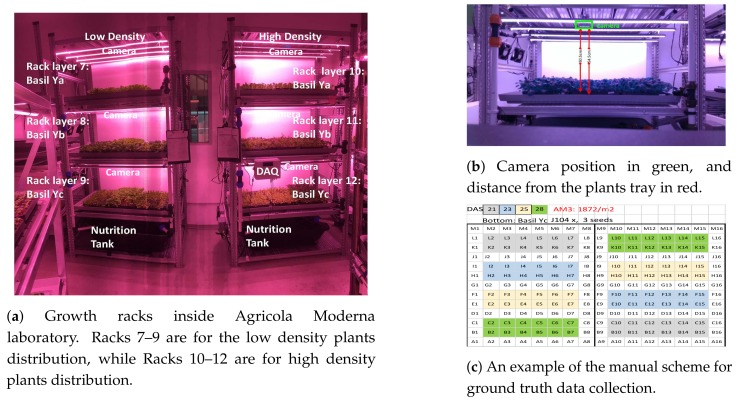
Experiments setting within the Agricola Moderna laboratory. (c) Each cell indicates a basil plant, for a high density tray collecting plugs of the Yc specie

**Figure 3 sensors-19-04378-f003:**
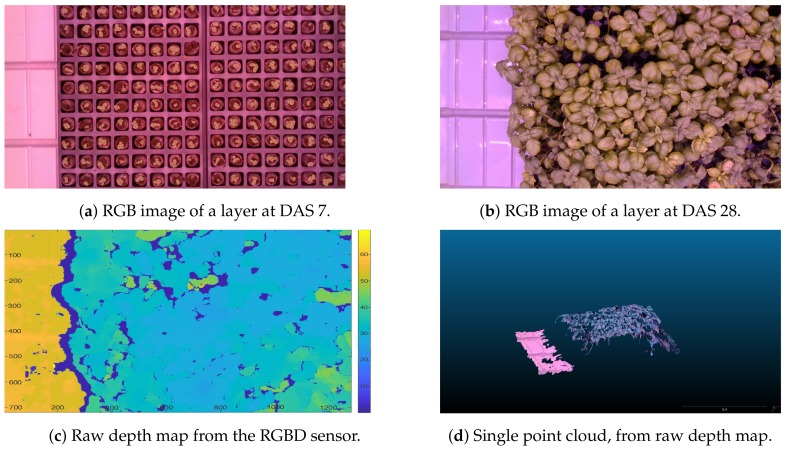
The data collected include 1728 color images and 1728 depth maps of the plants (better seen in color).

**Figure 4 sensors-19-04378-f004:**
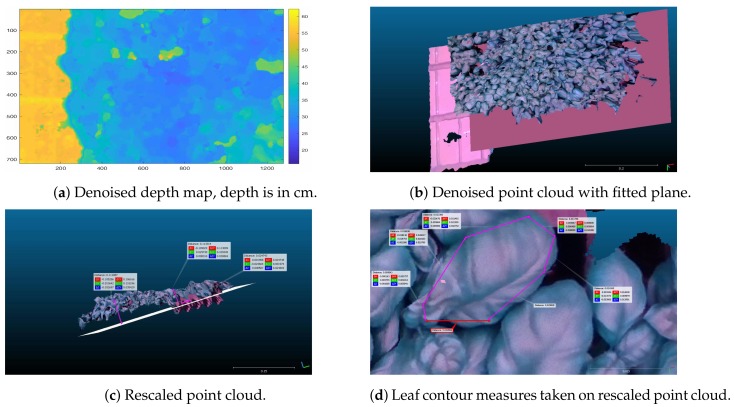
The figure shows relevant preprocessing steps for both depth map and point cloud.

**Figure 5 sensors-19-04378-f005:**
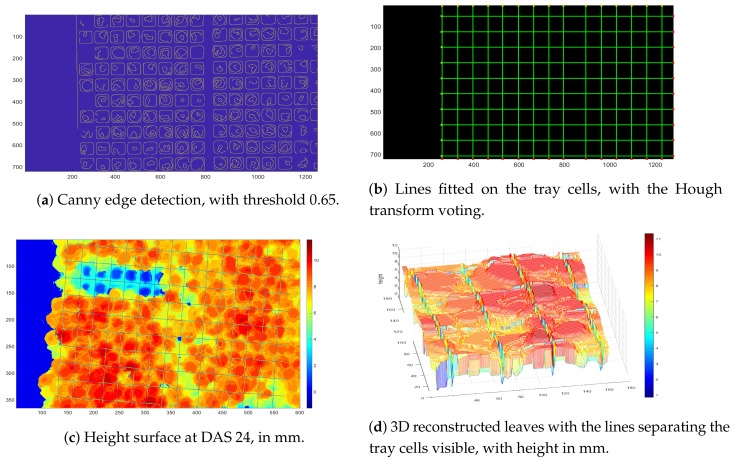
The figure shows the main steps to obtain the surface height of a tray partitioned according to the tray cells. (**c**) A section that has been cut out for measurements is visible.

**Figure 6 sensors-19-04378-f006:**
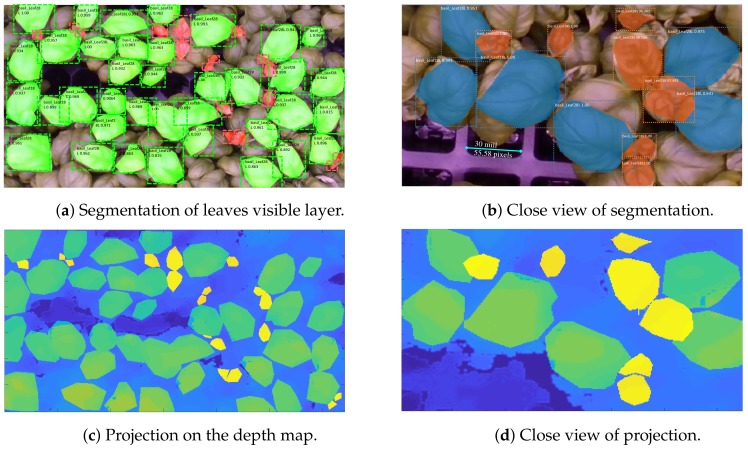
Segmentation of large and new leaves and projection of the segmentation on the depth map.

**Figure 7 sensors-19-04378-f007:**
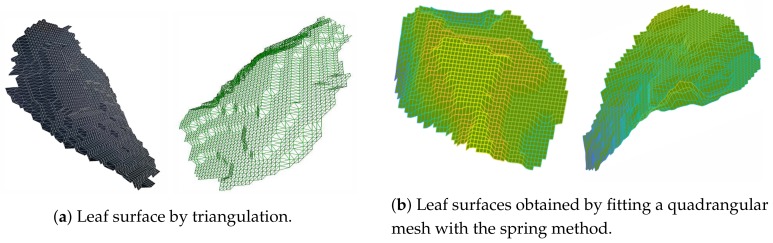
Examples of leaves surface geometry as obtained from the estimation process described in the paragraph.

**Figure 8 sensors-19-04378-f008:**
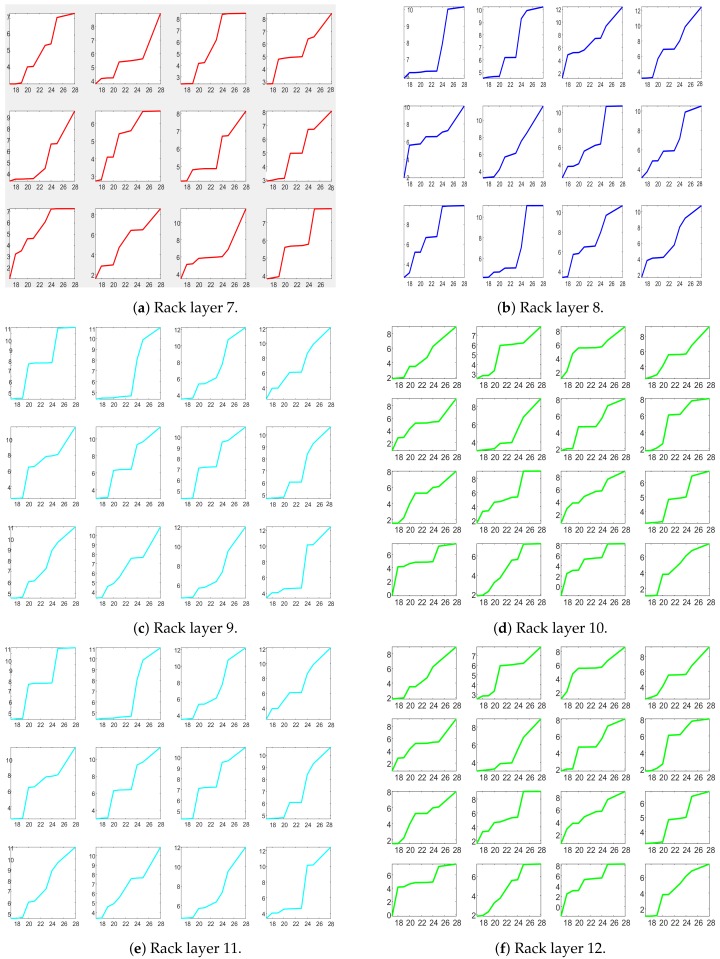
Plants height trends from DAS 17 to DAS 28 for groups of four cells in each rack layer, where cells were computed as described in Section 4.

**Figure 9 sensors-19-04378-f009:**
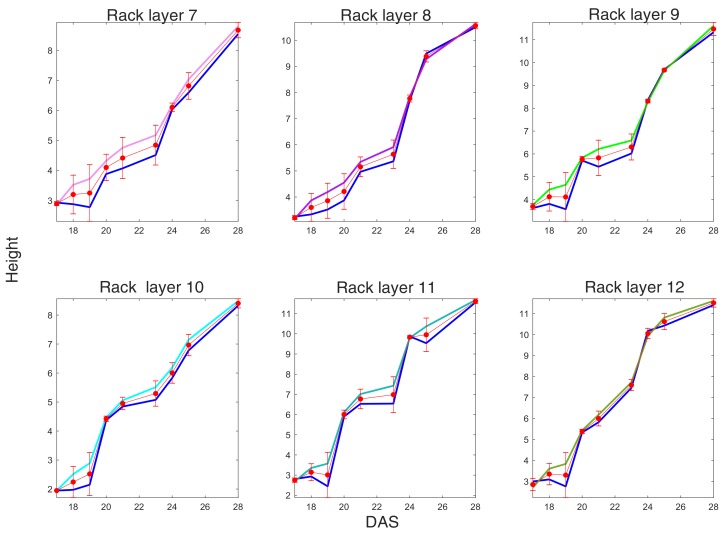
Plant height trends compared with ground truth, from DAS 17 to DAS 28 for Racks 7–12. Recall that Racks 7–9 have low density, while Racks 10–12 have high density plants distribution. For each graph, blue represents the ground truth height measurement taken manually, and the other colors the height predicted according to the vision based methods. Finally, the graph line in red, with red markers, and orange errors segments, shows the distance (see Equation (Equation 8)) between the ground truth and the predicted height.

**Figure 10 sensors-19-04378-f010:**
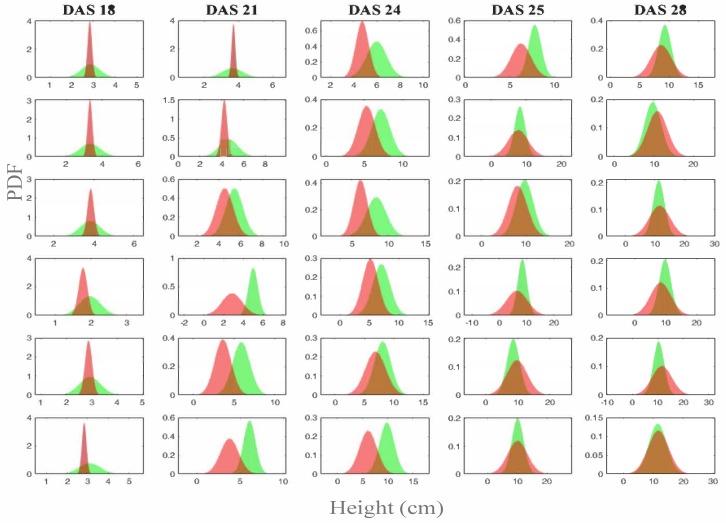
Plants height distribution for DAS 18, 21, 24, 25 and 28, and for Racks 7–12. Recall that Racks 7–9 are low density, while Racks 10–12 are high density. In green is the probability density function (pdf) generated by the mean and variance of the manually collected height measures, and in red is the pdf generated by the predicted mean and variance. Note that the pdf does not need to be up to 1. We preferred to give the pdf because it better highlights the increase of the variance as DAS increases, and it shows that it is also wider for low plant density racks.

**Figure 11 sensors-19-04378-f011:**
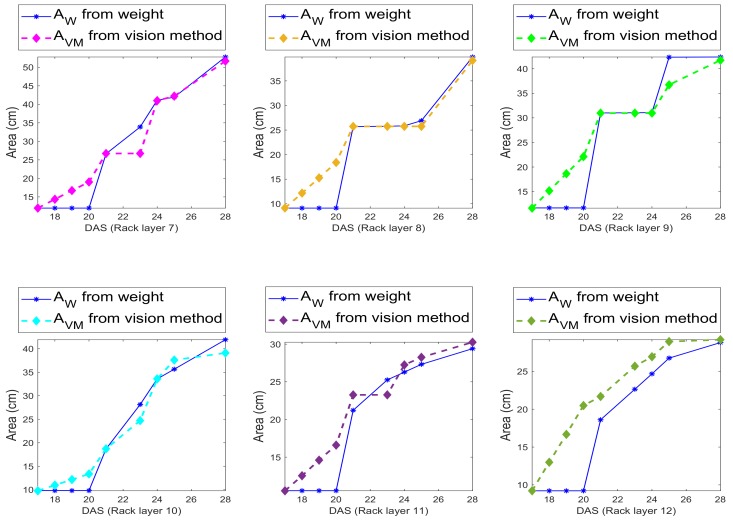
Leaf area trends for both the area computed by training a small set of samples, and by vision method, as described in Section 4.3. Data from vision are given for DAS 18–28, while the leaf area computed by the network, using the manually measured weights is only for DAS 21, 24, 25 and 28, for Racks 7–12.

**Figure 12 sensors-19-04378-f012:**
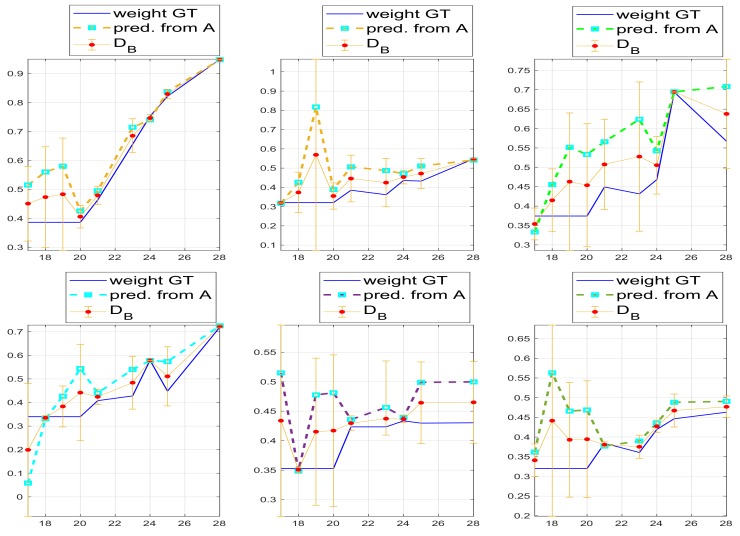
Leaf weight trends both computed from the predicted leaf area and as given by the manual measurements. As for plants height the error is computed basing on the Bhattacharyya distance between probability functions (see Equation (Equation 8)), for the same reasons, namely we have comparable probabilities, although not comparable samples. In fact, data from vision are given for DAS 18–28, while the area computed by the network, using the weights the leaf area, are limited to DAS 21, 24, 25 and 28, for Racks 7–12. We recall that Racks 7–9 (**top**) are low density plant distribution, while Racks 10–12 (**bottom**) are high density plants distribution.

**Figure 13 sensors-19-04378-f013:**
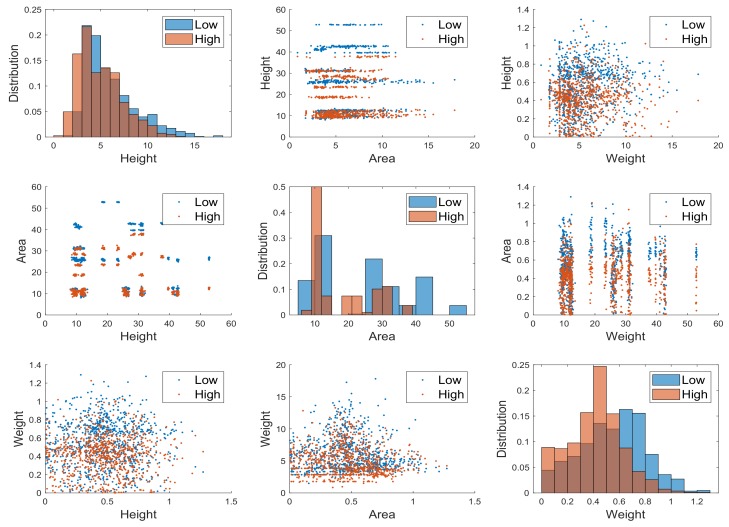
Features *plants height*, *leaf area* and *leaf weight* data as a pairwise scatter plot. The diagonal plots the marginal histograms of the three features. The off diagonals contain scatterplots of all possible pairs of features. As indicated in the legend, brown points are related to feature of plants gathered in low density racks, while the azure points are related to features of plants gathered in high density plants.

**Table 1 sensors-19-04378-t001:** Environmental Conditions

Racks	Low Density	High Density
	(Rack Layers: 7,8,9)	(Rack Layers: 10,11,12)
Density (plugs/m${}^{2}$)	624	849
Plugs per layer	104 × 2	144 × 2
Plants/Plug	3	3
Light cycle (hours/day)	16 and 20	16 and 20
Humidity (%)	65–70	65–70
Temperature (${}^{\circ }$C)	16 and 20	16 and 20
Light intensity (μ mol/m${}^{2}$)	200–240	200–240
PH	5.5–6.0	5.5–6.0

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
