# Peer review of "Vision Based Modeling of Plants Phenotyping in Vertical Farming under Artificial Lighting"

_sensors, 2019, doi:10.3390/s19204378_

Round 1
Reviewer 1 Report
The used the state-of-the-art vision-based methods for measuring the height and the leaf area. It needs minor revision.
Please describe the statistics in the abstract. How many data was measured? What does it mean that you have an entirely accurate result? Figure 1: label each figure in alphabetical order and describe each of them in the caption. Apply this technique for all the figures. Also, there is no need to say “the above figure” in caption of Figure 2, as the reader knows it refers to Fig 2. Moreover, the description of the figures is too much. For instance, you explained about the ICP and the fitted plane within the context and also below the figure. I am suggesting to just explain the topic in the content and put a few words as a label inside the caption. The content of some figures such as Fig 8 is not legible. This problem is almost detectable everywhere. Please revise the contribution section. No need to mention who did what. Also, change the future work paragraph to the end of the paper. The RGB-D camera achieves different accuracy level by changing the distance from the target. How did you determine to fix it at around 55 cm? What is the working ranges of this sensor? Also, first you described that you have multiple options for choosing the sensor, and the change your verb tense to singular without describing the reason for choosing that one. Please fix it.
Reviewer 2 Report
In this paper, a vision based method to predict the height of the plant and the leaf size and weight. It is useful for unman measurement.
1. In figure 7, the position of the RGBD camera should be labeled.
2. In section 4.1, how to get the four depth maps? I don’t understand how to use light flickering and ventilation get the four depth maps? And the four depth maps are synchronized?
3. In section 4.1, the point cloud is rescaled using a known metric in line 180. Why? The RGBD camera can get the scaled depth, why rescale the point cloud?
4. In section 4.2, the RGBD camera is the perspective model, not the orthographic model. So in figure top left of the figure 4 doesn’t represent the ground truth of the tray cell. I think you can use a homography transform to get the ground position if the tray cell can be regarded as a plane.
Reviewer 3 Report
This manuscript developed a novel method for vision based plants phenotyping in indoor vertical farming under artificial lighting. The technical content is, in my opinion, sound. However, I have some comments and suggestions for authors as below:
I think the Abstract was too simple and did not cover the main content. For example, RGBD imagery and how to predict the three phenotype features did not be mentioned. Abbreviations in this manuscript should be checked. For example, AI is first used in Line 40 and there is no full name.
“ML” was used in Line 58; however, “Machine Learning” was used in Line 64. Moreover, the capital letters for “Data Science”, “Computer Vision” and so on is unnecessary.
And some abbreviations that just used one time are unnecessary, for example LAI.
Maybe there was something wrong in “Volume and Leaf Area Index Calculation (VOL-LIAC)”. VOL-LAIC?
Figures 2-7 should add serial number such as (1), (2), and (3) or (a), (b), and (c) and so on for the subfigures. “Since each frame images 2 trays, each image collects either 144 * 2 plugs for the high density layers or 104 * 2 plugs for the low density layer”. Why 144 and 104? Especially why 2? Mask R-CNN was directly used? Maybe some detail should be introduced. The section “Conclusions” was needed.
Round 2
Reviewer 2 Report
The paper has been revised according to the suggestion, and I think it can be accepted.